# Contrast enhanced longitudinal changes observed in an experimental bleomycin-induced lung fibrosis rat model by radial DCE-MRI at 9.4T

René in 't Zandt[1]*, Irma Mahmutovic Persson[1,2], Marta Tibiletti[3], Karin von Wachenfeldt[4], Geoff J. M. Parker[3,5], Lars E. Olsson[2,6], on behalf of the TRISTAN Consortium[¶]

1 Faculty of Medicine, Lund University BioImaging Centre, Lund University, Lund, Sweden, 2 Department of Translational Medicine, Medical Radiation Physics, Lund University, Malmö, Sweden, 3 Bioxydyn Limited, St James Tower, Manchester, United Kingdom, 4 Truly Labs, Medicon Village, Lund, Sweden, 5 Department of Medical Physics and Biomedical Engineering, Centre for Medical Image Computing, University College London, London, United Kingdom, 6 Department of Hematology, Oncology and Radiation Physics, Skåne University Hospital, Malmö, Sweden

¶ Membership of the TRISTAN-IHI consortium (#IB4SD-116106) is provided in the acknowledgments.
* rene.in_t_zandt@med.lu.se

**Data Availability Statement:** All DCE-MRI data, image mask for the lung segmentation, histology

## Abstract

Identifying biomarkers in fibrotic lung disease is key for early anti-fibrotic intervention. Dynamic contrast-enhanced (DCE) MRI offers valuable perfusion-related insights in fibrosis but adapting human MRI methods to rodents poses challenges. Here, we explored these translational challenges for the inflammatory and fibrotic phase of a bleomycin lung injury model in rats. Eleven male Sprague-Dawley rats received a single intratracheal dose of bleomycin (1000iU), four control rats received saline. Imaging was performed on days 7 and 28 post-induction. Ultra-short echo time imaging was used to image the lung for 7 minutes after which Clariscan was injected intravenously. Lung signal changes were measured for an additional 21 minutes. Images were reconstructed with a sliding-window approach, providing a temporal resolution of 10 seconds per image. After imaging on day 28, animals were euthanized, and lungs were collected for histology. Bleomycin-exposed rats initially exhibited reduced body weight, recovering to control levels after 20 days. Lung volume increased in bleomycin animals from 4.4±0.9 ml in controls to 5.5±0.5 ml and 6.5±1.2 ml on day 7 and 28. DCE-MRI showed no change of initial gradient of relative enhancement in the curves between controls and bleomycin animals on day 7 and 28 post-induction. On day 7, the DCE-MRI washout phase in bleomycin animals had higher signals than the saline group and than observed at a later time point. Lung pixels were binned in 7 enhancement classes. On day 28, the size of low relative enhancement bins almost doubled in volume compared to controls and animals on day 7 post-induction. Histology on day 28 suggests that findings could be explained by changes in lung tissue density due to lung volume increase. Adapting this clinical MRI method to rodents at 9.4T remains a challenge. Future studies may benefit from lower field strength MRI combined with higher temporal resolution DCE-MRI.

sections and all tables to generate the figures are available from the Swedish National Data Service (SND) database (accession number https://doi.org/10.5878/b4cg-vc79).

**Funding:** The research leading to these results received funding from the Innovative Medicines Initiatives 2 Joint Undertaking under grant agreement No 116106 (IB4SD-TRISTAN). This Joint Undertaking receives support from the European Union's Horizon 2020 research and innovation programme and EFPIA. The funders had no role in study design, data collection and analysis, decision to publish, or preparation of the manuscript.

**Competing interests:** The authors have declared that no competing interests exist.

# Introduction

Formation of fibrotic tissue results from progressive fibrotic disease such as IPF, or as a consequence of lung injury that can lead to uncontrolled wound healing. Fibrosis may result in shortness of breath and chest pain [1]. While a cure remains elusive, anti-fibrotic treatments can help to slow down disease progression before it becomes too severe and impairs lung function [2]. Early detection and diagnosis are crucial for optimal patient care, making fast and non-invasive diagnostic tools, such as imaging, attractive. Non-invasive biomarkers that reflect the shift from inflammation to fibrosis may be particularly valuable, potentially enabling personalized treatment plans including early anti-fibrotic intervention. Animal models may provide an opportunity for identifying imaging biomarkers of use in this setting. In experimental research, the use of bleomycin as an inducing agent to model inflammation and fibrosis in animal models proves valuable due to its good repeatability [3–6].

Dynamic contrast enhanced magnetic resonance imaging (DCE-MRI) can provide longitudinal and regional information on lung function [7] and may be valuable for detecting the abnormalities in perfusion and capillary permeability known to occur in IPF [8]. Changes in the microvasculature and extravascular extracellular space have been detected previously in fibrosis using a model-free analysis of DCE-MRI contrast agent uptake curves [9]. In addition, a new potential imaging biomarker was discovered by Weatherley et al. by identifying changes in lung perfusion after analysis of first passage contrast agent characteristics in IPF patients [10].

The translation between the clinical setting and using DCE-MRI in mice is challenging. High magnetic field preclinical systems with high signal to noise capability only partly compensate for the required higher sensitivity that is necessary to image rodent lungs with similar anatomical detail as human lungs. Also, signal decay for lung tissue is much more rapid at the higher field strengths of preclinical systems than at typical clinical field strengths of 1.5- 3T. In addition, the high cardiac rate and short systemic circulation time in rodents puts an even higher demand on the temporal resolution required to obtain quantitative perfusion parameters. Acquisition acceleration techniques are widely available clinically but are commonly lacking in most preclinical MRI systems, increasing the technical challenge. It has been shown that a quantitative first passage DCE-MRI curve can be obtained in rat lung tissue in a single slice by repetitive contrast agent injections [11]. In human lung diseases, as well as in the bleomycin animal model, the disease is spatially heterogenous, and a single imaging slice may not be representative of the disease status of the lungs in general. Therefore, three-dimensional coverage of the lungs is preferred to achieve a better representation of the grade and distribution of the disease. However, three-dimensional acquisitions are slow relative to the circulation dynamics in rodent lungs currently not allowing measurement of a quantitative first passage DCE-MRI in 3D. Therefore, qualitative and model-free analysis methods are of interest, which do not focus solely on the first passage phase of the contrast agent dynamics and may be applicable to more slowly-sampled data [9].

The aim of this study was to explore the translational challenges of clinically used DCE-MRI methods in a fibrosis rodent model. An experimental bleomycin rat model was studied during its dominant inflammatory phase and in its later more fibrotic phase. Continuous radial multi-slice ultrashort echo time (UTE) imaging using sliding window reconstruction was employed to enable optimization of acquisition temporal resolution versus volume coverage and spatial resolution [12]. Our hypothesis was that the characteristics of the contrast enhancement time curves could be linked to the inflammatory and fibrotic stage of the bleomycin animal model.

## Material and methods

### Animals and statement of ethical approval

*Sprague-Dawley* male rats (n = 15) were purchased from Janvier Labs (Le Genest-Saint-Isle, France). The rats were nine weeks old at the start of the experiment and were housed at Lund University animal facilities with 12 h light/dark cycles and fed *ad libitum*. All animal studies were ethically reviewed and carried out in accordance with European Directive 2010/63/EEC and reported according to the ARRIVE guidelines [13]. The studies were approved by the local ethical committee in Lund/Malmö, Sweden, before they were initiated (ethical permit number 4003/2017).

### Bleomycin challenge

One single intratracheal dose of bleomycin (Sigma Aldrich, St. Louis, MO, USA), was administered on day 0 (n = 11), with a concentration of 1000 iU, dissolved in 200 μL saline. Control animals (n = 4) received the same volume of saline. Intratracheal administration was performed as previously described [4]. Based on prior experiences with the bleomycin model, 5 out of 11 animals were designated as backup subjects for the MR imaging group of bleomycin exposed animals, to maintain the study's reliability and continuity in case of unexpected issues with the primary subjects, such as extensive weight loss or imaging or anesthesia related events. Day 7 and day 28 were chosen for DCE-MRI scan days as two representative days of dominant inflammatory and fibrotic time points respectively [4,5]. The planned group size for DCE-MRI in bleomycin exposed animals was n = 6.

### Injection of the contrast agent

Prior to the scan sessions, a stock of contrast agent was prepared at a concentration of 0.15 mmol/ml (Clariscan, GE Healthcare, UK). The injection site was prepared using a Venflon catheter (BD Venflon™ Pro, Becton Dickinson Infusion Therapy) by flushing with Heparin (Heparin, LEO Pharma) dissolved in saline to keep the injection site open and prevent coagulation. Subsequently, 1 ml contrast agent was injected in each rat intravenously via the tail vain. The injection was carried out manually in the magnet room with an estimated injection time of 5 s.

### MRI

MRI experiments were carried out using a 9.4T Bruker Avance III spectrometer running Paravision 7.0 (Bruker, Ettlingen, Germany) using an 86 mm diameter quadrature body radiofrequency (RF) coil. The rats were initially anesthetized in a box using a gas mixture of $O_2$ and $N_2O$ (1:1) with 3% Isoflurane (IsoFlo Vet, Orion Pharma Animal Health). Once the animal was sedated and moved to the preparation table, the anaesthesia was maintained at the same gas flow but with reduced Isoflurane at 1.8–2.0%, delivered via a nose cone, while the catheter was inserted in place. Subsequently, the animal was moved to the MRI system with continuous isoflurane supply. During the entire scan session, monitoring of the respiration and the body temperature of the animal was done using a rectal temperature monitoring probe and a pneumatic pillow for tracking breathing (SA Instruments, USA). The Isoflurane gas was manually adjusted to maintain a respiration rate of 60–80 breaths per minute. Animals were kept warm using a warm water bath (Lauda RC6, Lauda-Königshofen, Germany) connected to tubes built in the animal bed. The rats were then imaged applying a low-resolution respiration triggered scan to confirm optimal positioning of the animal in the magnet bore followed by a higher-resolution respiration triggered coronal multi-slice scan covering the lung. The latter image set

was used for the anatomical positioning of the UTE images. The Bruker 2D-UTE sequence code was modified to allow for the number of radial profiles to be oversampled for continuous center out scanning [12]. Further, the gap between subsequence radial profiles was changed to a tiny golden angle of 23.63 degrees [14]. Further experimental details: echo time (TE) = 0.368 ms, 38 slices with a thickness of 1.208 mm, in plane resolution of $0.604/0.604$ mm$^2$ during acquisition and a field of view of 58x58 mm$^2$. The repetition time (TR) per radial profile was 3 ms. The looping structure in the Bruker UTE sequence code acquires spokes for all slices at each spoke angle, before acquisition of the next angle is started. The effective repetition time for each slice was 114 ms (38 slices, TR = 3ms). To achieve good T1-weighting for sensitivity to the contrast agent, the excitation flip angle was set to 40 degrees which is a compromise between T1-weighting, hardware heating and excessive RF power deposition for the 28 minutes of continuous high duty cycle scanning. The receiver gain was kept fixed at 64 dB for all scans. After baseline scanning of 7 minutes, the contrast agent was injected manually. Scanning continued for an additional 21 minutes to acquire data for a total of 28 minutes.

Although the receiver gain settings were kept fixed for all animals, the increase in weight between the scanning dates changes the load of the RF coil and could potentially had influenced the final image quality. As a reference check, the signal to noise ratio (SNR) of the images for every animal was estimated based on the noise and signal level of a slice around the spinal cord from the respiration triggered coronal scans (non-radial scanning). Further, the reference power level in Watts required by the system was noted.

## Image reconstruction

The Bruker software platform reconstructed the (average) contrast enhanced image representing the signal measured over the whole 28 minute UTE scan time. This data volume was exported to images in Dicom format for further inclusion in the analysis pipeline. The UTE raw data was processed using a home-written Matlab script (Matlab R2022b, Mathworks). The script uses the pvmatlab library (Bruker Biospin, Germany) for the convenient import of Bruker raw data into Matlab. Both the pre-scans for the measured trajectory and the B0 trajectory calibration were imported. The phase evolution in the B0-file was added to the unwrapped phase of the raw data. The reconstruction of the images from the radially acquired data was achieved by regridding the k-space points to a Cartesian coordinate system using the Bart toolbox [15,16] by applying the following steps: the normalized trajectory values from the Bruker (scaled between -0.5 and 0.5) were multiplied by the size of the acquisition matrix (96), to get the correct trajectory scaling format required by the Bart toolbox. All radial profiles were then weighted in the regridding process by the inverse of the radial sample density given by the normalized magnitude of the 2D-trajectory matrix. Subsequently, the k-space raw data was reorganized using the sliding window concept. Image 1 in this array was reconstructed using radial profiles 1–90, while the second image was reconstructed using profiles 60–150, etc. The reconstructed images used data acquired over 90 spokes times the effective repetition time of 114 ms, which equals 10.2 s and a step size of 60 profiles which equals to 6.8 s. The final array size of 247 sliding window DCE-frames reflected the highest achievable time resolution per image in our setup.

The regridded Cartesian k-space data matrix, containing all radial profiles, all time points and all slices as well as the trajectory matrix, was used by the <pics> command in Bart toolbox v0.0.8 [15]. Empirically determined l1 (0.00536) and l2 (0.0016) regularization constraints with a ratio l1/l2 = 3.35 [17] were used to reconstruct the images using zero-filling with a factor of 2, e.g., 192 x 192 pixels and an apparent pixel resolution of 0.302x0.302 mm$^2$. After reconstruction, the images were exported in Dicom format.

To determine the lesion size using histogram analysis of the pixel intensity distribution in the lung as described previously [4], a higher quality baseline image was used that was obtained by reconstructing the baseline image using 3000 spokes. The procedure of the reconstruction of these baseline images was as described above for the high time resolution reconstruction.

## DCE analysis

DCE enhancement was analyzed with a model-free analysis method using MiceToolKit (MTK 2022.4.9; NONPI Medical AB, Sweden). Lung tissue was segmented manually per slice using the drawing tools available in MiceToolKit following the previously used criteria of including both large vessels and high-signal intensity areas while heart tissue was excluded [4,5]. The total lung volume was calculated by the number of pixels in the segmented region of interest (ROI) and multiplied by the pixel volume. The lesion size was calculated by using the higher quality baseline images and the volume of the lesion size was plotted for control and bleomycin animals on day 7 and day 28.

The relative normalized enhancement curves for the whole lung were calculated by plotting the average relative enhancement for the lung ROI as a function of time. The baseline signal was calculated by averaging image frame 1 to 50 for each animal. For every time point the relative signal enhancement was calculated by normalizing signal change to the baseline signal and plotted groupwise, e.g., control, day 7 bleomycin and day 28 in the bleomycin group.

The average signal curve was dominated by the signal in the vessels in the lung. To visualize the dynamics of the contrast enhancement in lung tissue with different signal intensity, the pixels were binned using 6 different intervals for the relative signal enhancement ratios to create 7 different clusters. These 6 intervals were based on the maximum relative signal enhancement measured directly after the injection of the contrast agent (intervals used for the relative signal enhancement 1, 2, 3, 4, 8 and 12). The background signal of the images was measured outside the object in the periphery of the UTE image and its normalized signal change to the baseline background was calculated for every time point. These time-sensitive relative changes of the background signal were subtracted from all DCE curves on a per animal base prior to further analysis. Thereafter, the average curve for each bin was plotted for the three animal groups. From the number of pixels in every bin, the volume represented by the 7 different clusters was calculated for each animal group.

## Histology

After the last scan session, at day 28 post-induction, all 15 rats were terminated by intraperitoneal overdose of PentobarbitalSodium (Apotek Produktion & Laboratorier AB, Stockholm, Sweden). The lungs were insufflated with fixative solution using 4% paraformaldehyde introduced via the trachea then dissected and submerged in fixative solution, until further processing. Subsequently, the lungs were embedded in paraffin and sectioned at four different positions in the sagittal plane, to 4 μm thin sections. For histological analysis lung sections were stained using hematoxylin and eosin (H&E; Histolab Produts), Masson's Trichrome staining kit (Polysciences, Hirschberg and der Bergstrasse, Germany) and Picro Sirius Red staining kit (Abcam). After quality assessment of the tissue sections, confirmation of fibrotic foci formation was found within the lung at all investigated positions (Section I-IV). Thereafter, the Masson's trichrome stained sections were scanned (Zeiss Axio scan Z1 imager, Carl Zeiss microscopy GmbH, Germany), and tissue/air ratio—density was quantified. First, the digitized microscope images were exported to JPEG using a quality of 95% and a resize value of 10% (Zen blue 3.7, Carl Zeiss microscopy GmbH, Germany) to reduce the file size. The JPEG images were imported in MiceToolKit, where the tissue was manually segmented using

the drawing tools available. The tissue density was estimated by the volumes calculated by considering all pixel values in the sectioned image below the experimentally determined value of 900 as air and all above 900 as lung tissue. The tissue fraction was calculated in all sections (I-IV) for each lung.

## Statistical analysis

All data were analyzed using software GraphPad Prism (v 10.1.2; GraphPad Software, San Diego, CA, USA). Data were expressed as mean values and standard error of the mean (SEM) unless otherwise specified. DCE curves were compared using a two-way ANOVA. Two sided unpaired parametric t-tests were used except for the lung volume where a one-sided parametric t-test was applied. P-values of less than 0.05 were considered statistically significant. Significance was indicated by * when $p < 0.05$; $p < 0.01$ by **; when comparing bleomycin-exposed group to the corresponding controls.

## Results

The bleomycin challenge induced weight loss, followed by a recovery phase. Between days 21–28 the bleomycin-exposed animals recovered and had similar weight as controls (Fig 1A). The lung volume increased in the bleomycin-exposed animals from 4.4±0.9 ml to 5.5±0.5 ml on day 7 (p<0.05) and to 6.5±1.2 ml on day 28 (p<0.01, Fig 1B). The volume of the high intensity pixels from the histogram analysis pre-contrast, showed an increase from 0.64±0.10 ml in controls to 1.03±0.30 ml on day 7 (p<0.05) and 1.00±0.28 ml (p<0.05) on day28 in the bleomycin-exposed animals (Fig 1C). The increase in weight affected the required reference power levels of the MRI system, but the SNR was constant both for control animals and bleomycin-exposed animals on day 7 and day 28 (S1 Fig).

DCE-MRI was successful in 3 out of 4 control animals on day 7, for one animal the Venflon catheter punctured the venous artery and substantial amount of fluid was injected extra-vascular in the tail. This animal was removed from the DCE-MRI on day 28 due to vascular damage in the tail. Subsequently, only 3 controls were available for DCE-MRI on day 28 as well. In the bleomycin-exposed group, all DCE-MRI injections (n = 6) were completed successfully on day 7. On day 28, one contrast agent injection partly failed as the images showed that only a fraction of the contrast agent reached the lung. This reduced the group size for DCE-MRI in the bleomycin-exposed group to n = 5. The bleomycin-challenged animal group presented lesions in the lungs in the baseline UTE images (Fig 2A) although lesions were not well distinguished from vasculature. The initial relative DCE enhancement between the bleomycin group versus

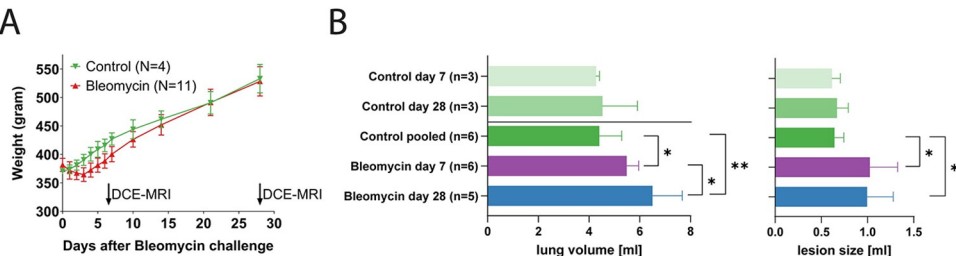

**Fig 1. Lung volume and animal weight in bleomycin exposed animals versus controls.** A. The bleomycin-challenged animals lost about 10% of their body weight after a few days and recovered during the next weeks. After day 20 post-induction, no significant difference in body weight was observed. B. The average lung volume measured by MRI increased significantly in the bleomycin-challenged lungs at day 7 and 28 compared to controls. C. The average volume of the high intensity pixels increased in the bleomycin-exposed lungs and is considered to be a marker for the lesion size.

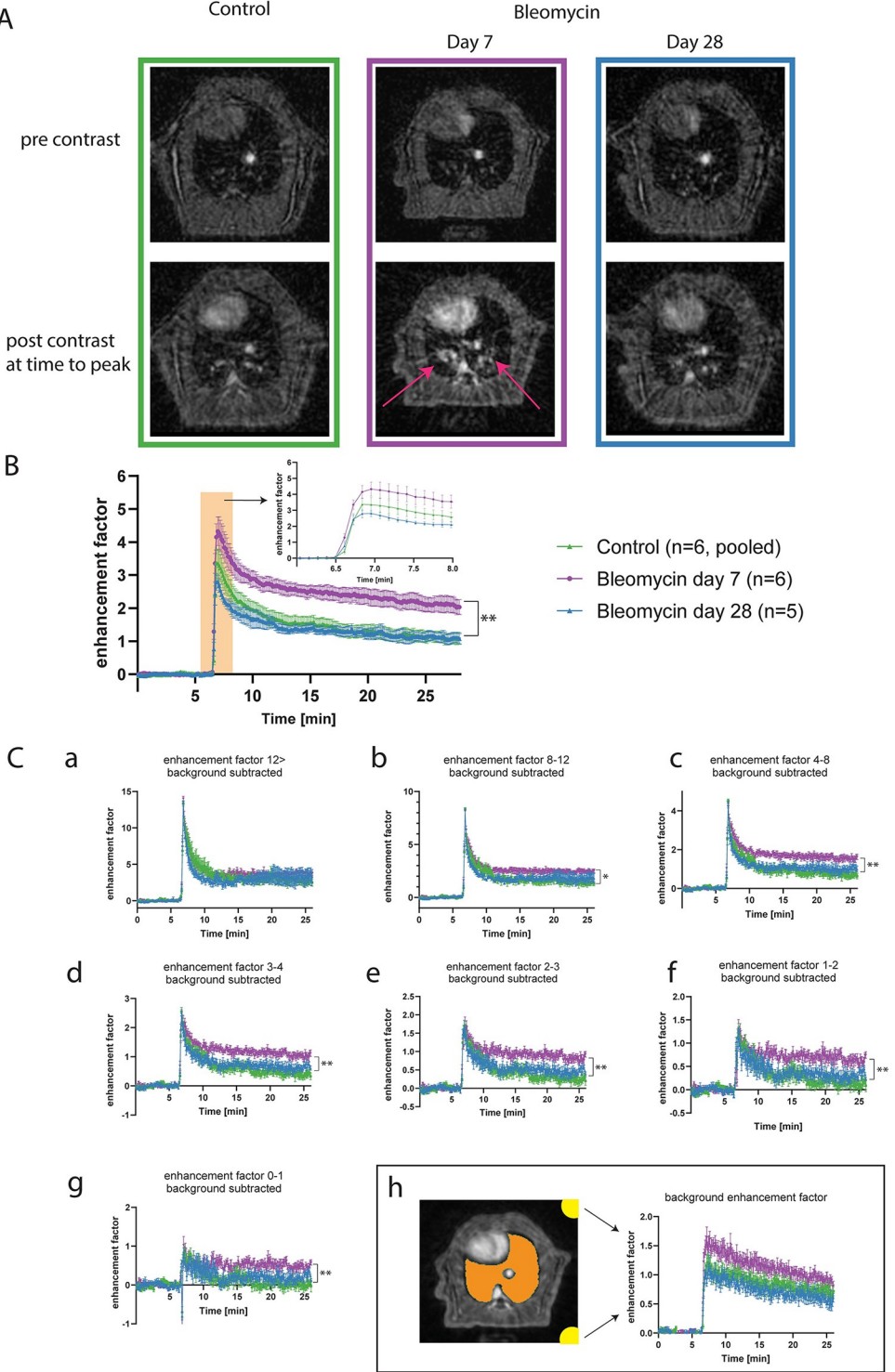

**Fig 2. Multi-slice DCE MR images at a frame rate of one image every 10.2 seconds with a step size of 6.8 seconds.**
A. Representative pre- and postcontrast images are shown for control, and animals on day 7 and 28 post-induction with bleomycin, at one slice location of the 3D dataset. The relative signal enhancement by the contrast agent could be observed in all animals. Lesions were more difficult to depict due to the large vascular signal in the lung region. The red arrow points out the lesions at 7 days. B. The group average of total lung signal enhancement as a function of time was plotted for the bleomycin-challenged animals and controls and showed similar enhancement after injection and a significantly sustained higher signal for the bleomycin group at day 7 (p<0.01). C. DCE-curves for 7 different relative

enhancement signal level intervals. The background signal composed of artefacts and noise was measured within the yellow ROIs (C.h). This signal was subtracted, per animal prior to averaging, from the curves shown in C.a-g.

control was found to be indistinguishable in the initial phase directly after the injection of the contrast agent (Fig 2B). The relative enhancement curve at day 7 was significantly different between the bleomycin animals (p<0.01) and controls, with a sustained higher average relative enhancement for the bleomycin animals after injection of the contrast agent.

Radial artefacts and noise level were not constant during the acquisition, as demonstrated by the signal measured within the yellow labeled ROIs in the corner of the image outside the animal (Fig 2C.h). This normalized relative and time-dependent background signal was subtracted from all the relative enhancements curves for the 7 different signal intervals (Fig 2C.a–g) and were corrected for each animal. The DCE curves for the different relative signal intensity intervals revealed that the upslope of the initial average enhancement was not different between the bins or between the animal groups as data points were overlapping.

The average dynamic enhancement curves for the category representing the highest signals were not different between the animal groups in contrast to the average signal over the whole lung without binning (Fig 2C.a). The DCE further demonstrated that the bleomycin-exposed animals imaged on day 7 post-induction of disease all showed a sustained higher signal level during the washout phase for all other remaining bins. No detectable difference was found between the bleomycin group 28 days post-induction and controls for any bin.

The anatomical position of the pixels in the lung representing the tissue remodeling and lesion, and the different signal intensity intervals are shown for two different slice positions and for three animals (Fig 3). The subjects were chosen based on being most representative of the total group average. On day 28, the bleomycin-exposed animals had a significantly higher volume (p<0.01) of low enhancing pixels of 4.8±0.6 ml compared to 2.7±0.3 ml and 2.7±0.2 ml in controls and the bleomycin group on day 7, respectively, as calculated by summation of the volume represented by the three lowest signal bins, i.e. bins 0–3 (Fig 3B). The fraction of the total lung volume with a relative signal enhancement less than 3 times of the baseline level was 74% (4.8 ml with 6.5 ml average total lung volume) in bleomycin-exposed animals at day 28 post-induction compared to 61% in control animals (2.7 ml on a 4.4 ml average total lung volume). This fraction was found to be 50% (2.7 ml on a total lung volume of 5.5 ml) in bleomycin-exposed animals on day 7 post-induction.

Histology sections stained by Picro Sirius Red staining, in animals on day 28 post-induction confirmed that the bleomycin model induced substantial collagen production and accumulation in the lungs of these animals (S2 Fig). The tissue to air fraction was estimated based on microscopy sections in 4 control animals of which 3 were scanned using DCE-MRI and for 11 bleomycin-exposed animals of which 5 were scanned on day 28 using DCE-MRI (Fig 4A). During inspection of the samples, it was noted that the inflation procedure during fixation of the lung tissue clearly failed for one of the control animals and for one of the bleomycin-exposed animals. This reduced the size of the bleomycin group to 10 individuals and the control group to 3 animals for the tissue fraction quantification. The tissue fraction showed an increasing trend between the control animals and the bleomycin group for the tissue slice position through the left lung lobes (I-IV) and reached significance in the lateral region in section IV. The tissue fraction decreased from 0.43±0.05 to 0.33±0.06 (Fig 4B). This corresponded to an increase in the air volume of 30% in this region.

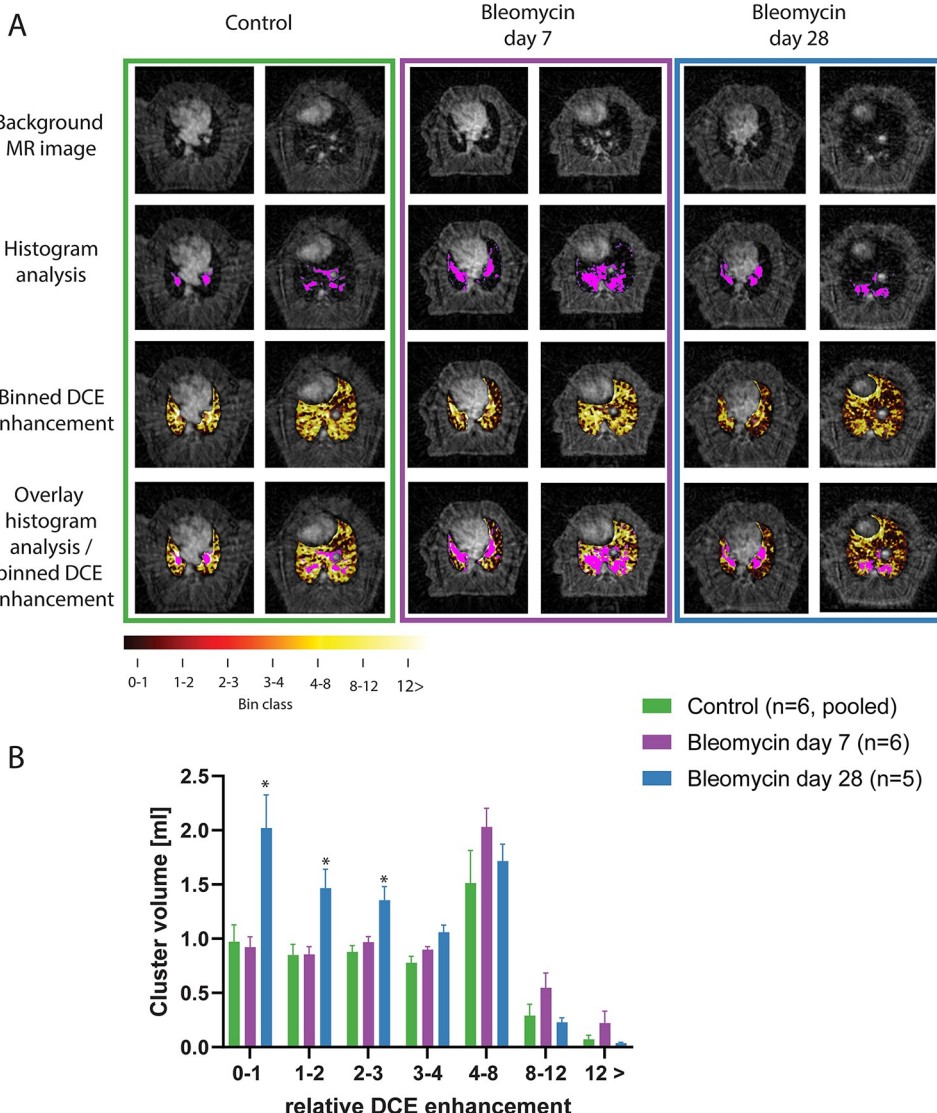

**Fig 3. Relative signal enhancement cluster location in the lung and its volume.** A. MR images of the lung shown at two representative locations. The images chosen had findings that were most representative of the average bin distribution of the group. The size of the area of tissue remodeling and lesion are overlayed onto the images in pink. The location of the pixels for the 7 binning classes was colocalized on the anatomical MR images for the bleomycin-challenged animals on day 7 and 28 versus controls. Finally, both colormaps were shown simultaneously on the MR images. B. The average group size in milliliters of the 7 bins were plotted for controls and the bleomycin groups on day 7 and day 28. On day 28 after bleomycin induction, the animals showed a large portion of the lung with low signal enhancement (relative enhancement 3 and lower).

## Discussion

The evolution of animal weight, the changes of the lung volume in bleomycin-exposed animals compared to the controls and the increase of the lesion size, all follow a similar pattern as observed previously in this model [4,5,18]. DCE-MRI in the experimental bleomycin animal model clearly identifies the inflammatory phase of this model around day 7 by a higher average signal enhancement over the whole lung as well as by grouping the relative enhancement from different relative enhancement signals. The curves representing the highest signal enhancement are not different between the groups, and likely, this segment represents the pixels

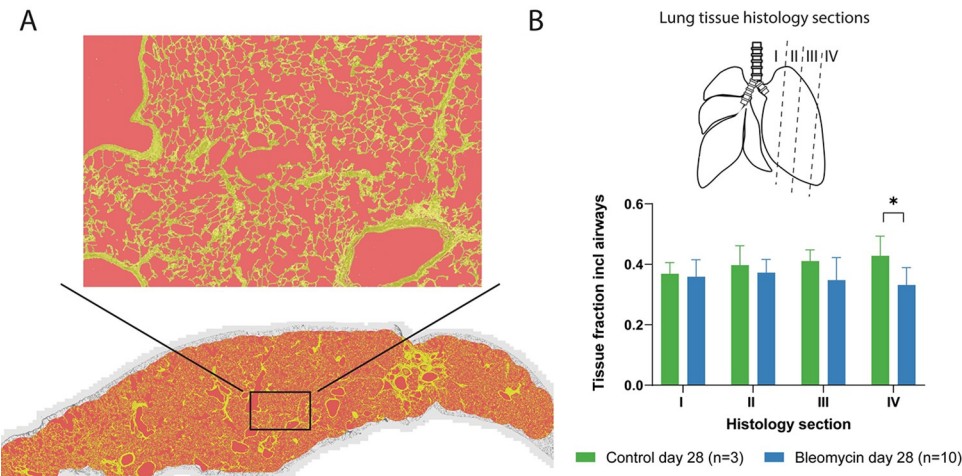

**Fig 4. Tissue fraction estimation of lung histology sections.** A. Concept for the estimation of tissue fraction. Yellow pixels in the image were considered to represent tissue without any specification on cell- or tissue type, while the red pixels inside the region of interest were considered to represent air. B. Histology sections I-IV through the lung were analyzed and the tissue fraction was plotted as a function of the location through the lung. Data presented from n = 3 controls and n = 10 bleomycin exposed animals.

originating from the large vessels only. Although significant changes in the area representing tissue remodeling and lesions have been shown before [4,18], our data failed to show a significant decrease in lesion size between day 7 and day 28 post bleomycin induction. The group size might be a limitation to reveal significant differences, especially considering that animals can show different amount of disease severity such as low and high responders towards bleomycin exposure [4].

The fibrotic stage of the experimental bleomycin model around day 28 is characterized by an increased lung volume and by 75% of the lung lacking significant relative enhancement after injection of the contrast agent. Neither the rate of enhancement nor the washout rate was found to be different in any lung region for bleomycin-exposed animals and controls based on the binning of the DCE signal enhancement at this time point.

The fast enhancement and lack of differences in the rate of MRI signal enhancement, or washout, suggests that the DCE signal measured directly after injection of the contrast agent represents either vascular signal or the leakage of contrast agent from the vasculature into the neighboring tissue with limited extracellular components. It is known from literature that the experimental bleomycin model induces increase in the overall mean lung density on CT measurements [19]. Changes in lung tissue density are a confounding factor when interpreting DCE signal enhancement based on the difference signal before and after injection of the contrast agent and therefore the DCE curves are preferable expressed by the relative enhancement to normalize for the lung tissue density differences [20]. Increase in collagen content could potentially lead to a change in dynamics of the relative contrast enhancement curves. However, the curves do not differ in shape for any location in the lung, both in controls and bleomycin-exposed animals in this study. Further, collagen synthesis has been correlated to colocalize with the area for tissue remodeling [4,18]. These regions are located centrally in the lung and coincide with the location of larger blood vessels. However, changes in the areas of the lung that have a low MRI signal and with less enhancement after contrast agent injection are located more laterally in the lung in our study.

While collagen formation leads to an increase in lung density in the experimental bleomycin model, the bleomycin challenge also results in an increase in lung volume that appears to result in some regions with a lower tissue density. In these regions, the DCE signal is even more challenging to analyze in our experimental settings as mentioned above. The increase of total lung volume of 2.1±0.5 ml for bleomycin-exposed animals on day 28, resembles the size increase of the lung areas characterized by the three lowest bins for the relative signal enhancement, i.e. 2.1±0.7 ml. This suggests that the expansion of the lung is mostly located towards the peripheral segments of the lung, as the lowest enhancing pixels were found in these areas. This is also confirmed by the decrease of tissue density based on the histological analysis represented by the lung sections assessed at four different positions throughout the left lung lobe. These observations can be linked to the compensatory inflation and stretch of the lungs, to overcome the lesion-contributing limitations of gas-exchange in the bleomycin-exposed animals, as observed previously in this model [4,5,19,21].

Histology confirmed the formation of collagen after 28 days and most collagen structures were situated near the larger airways as a direct consequence of the intratracheal administration of the bleomycin [4,18,19]. The hypothesis is that slow or delayed uptake of contrast agent into tissue would occur due to increased collagen content, which we did not find. However, if present at all, those areas would be difficult to detect as they are likely masked by the high signal intensity areas in the central region of the lung arising from the vessels. Furthermore, even when radial MRI data acquisition is less sensitive to motion artefacts in the MR images, substantial partial volume effects are present in the DCE-MRI data as the data was not synchronized with neither the respiration nor the cardiac movement during the acquisition in this study. Finally, collagen rich tissue is known to require short echo times to be detectable with rodent MRI at higher field, typically shorter than around 0.5 ms [4,18]. As these collagen rich regions coincide with high vascular signals, it cannot be excluded that high levels of paramagnetic contrast agents, such as Clariscan, will locally shorten the T2/T2*-relaxation time [22,23] even more, rendering the collagen rich areas invisible in the DCE-MRI experiment performed.

Physiologic motion of the heart and lung will contribute to streaking artefacts in the image during radial scanning especially with the minimal number of spokes used in the reconstruction to achieve the high time resolution that is required. Signal increase by injection of contrast agent will further amplify these artefacts and the effect is manifested as an increase in the global background signal of the whole image. Without correction for the background signal, a major part of the enhancement in the lung will be significantly biased by this background leading to misleading interpretation of the DCE enhancement. Furthermore, the movement of the lung in the slice direction, when the lung compresses and expands, contributes to partial volume effects. This could contribute to the reduced signal in the inferior lobe.

In our previous work using the bleomycin model, the lesion size in the lungs was identified using histogram analysis of the signal distribution of the anatomical MR lung image [4]. The volume of the lesions reflecting the inflammatory areas in the lung, was estimated to increase with around 1 ml for the bleomycin-exposed animals at day 7 post-induction [4] while in this study around 0.5 ml increase was found. Differences in image acquisition and image contrast might be the reason for this. The volume increase of pixels representing a relative enhancement of larger than 4 in bleomycin-exposed animals scanned at 7 days post-induction did not reach significance compared to control animals. However, the overall mean relative enhancement over the whole lung was clearly higher at the 7-day timepoint post-induction likely reflecting the inflammatory state of the lung tissue. Also in previous work, the tissue signal volume on day 28 in the bleomycin group, interpreted as fibrotic tissue, was estimated to be around 0.2 ml based on the difference in signal from the UTE images at two different echo times [5]. These pixels were all located centrally and overlapping with the area calculated using

histogram analysis. These pixels also corresponded to the tissue location associated with excessive collagen production based on histology [4,18].

Detection of collagen rich lung tissue could not be shown in our study using DCE-MRI in the rat lung. In the same animal model, PET scanning has been performed by using a specific collagen-I tracer, where collagen formation was clearly shown in the central part of the rat lung [5]. PET imaging has a several orders of magnitude higher sensitivity than MRI, and in the case of collagen-specific targeting radio tracer, no signal except for the collagen rich area is easier found. Other approaches have focused on the change in T2-relaxation values observed in the rat lung after the bleomycin challenge [24,25]. The T2-relaxation time increased in those tissue regions with inflammation in agreement with the lesion area detected by UTE-MRI at a longer echo time [4,24]. T2-mapping performed at a 2.0T magnetic field strength and by using a ventilator could clearly correlate both the proton density and T2 changes during the inflammatory and the fibrotic stage [25].

The animals in this study were monitored over 4 weeks of time and reached a weight of about 500 grams. The size of the animals at day 28 required the use of the largest volume coil (86 mm diameter, Bruker Biospin, Germany) available in the MR facility, and this was used throughout the entire study. A slightly better image quality can be achieved by the 72 mm body coil (Bruker Biospin, Germany) but this setup only works optimally for animals weighing up to 400 grams. A dedicated chest coil for larger rats would be beneficial for a better SNR as well as a limited volume of RF excitation limiting the artefacts in the UTE.

Substantial inflow effects and the relative low time resolution compared to the mean transit time in the rat lung [11], hampers a model-based analysis in our experiments. Additionally, these factors complicate the direct translation of perfusion-sensitive DCE-MRI used clinically, as this application relies on the measurement of the first-passage characteristics of the injected contrast agent [9,10]. It cannot be excluded that using a scanning DCE approach with significantly higher temporal and spatial resolution more details in the bleomycin model could be revealed. However, a large portion of the lung is lacking significant DCE-enhancement due to the challenges when scanning at 9.4T and a significantly higher signal to noise level in the images are probably required as well.

Translation of the DCE-MRI approach from the human setting into rodents has shown to be challenging. MRI has been implemented in a rodent model of fibrosis and the changes in vascular permeability during the inflammatory phase at day 7 has been demonstrated. It is currently not clear if the increase in low enhancing tissue on day 28 is correlated to the increase in fibrotic tissue, changes in vascular permeability or due to lung volume changes. Further studies in rodents should preferably be carried out using MRI systems with substantially lower field strengths for more beneficial T2-relaxation times in lung imaging and improved signal to noise ratio as a result.

## Supporting information

**S1 Fig. Signal to noise as a function of animal weight.** Increase in animal weight will change the load of the RF coil and potentially leads to changes in image quality. To establish the signal to noise over time, the noise level and the signal level were measured in an image used for localization purposes. Here, conventional Cartesian scanning was employed, after which the signal to noise ratio was calculated. Further, the power levels needed for the scanning established by the MRI scanner were plotted as well. Clearly, more power was needed for the animals on day 28, but the signal to noise ratio of the images was not different between the animal groups and time points.
(TIF)

**S2 Fig. Histology sections showing the presence of collagen.** Lung tissue sections were stained using Picro Sirus Red staining kit (Collagen: Red, Muscle Fibers: Yellow, Cytoplasm: Yellow). Clearly, the presence of collagen can be visualized, and heterogeneity is depicted for different disease severity and locations in the lung.
(TIF)

## Acknowledgments

Lund University BioImaging Centre (LBIC), Lund University, is gratefully acknowledged for providing experimental resources. Special thanks to Susanne Strömblad from LBICs histology platform for skilled tissue sectioning. TRISTAN-IHI consortium (#IB4SD-116106, https://www.imi-tristan.eu/) (Translational Imaging in Drug Safety Assessment—Innovative Health Initiative) consists of the following members: EORTC (Coordinator, International non-profit organisation); Pharma Companies Abbvie, Bayer (Lead), GSK (Co-lead), Merck Sharp & Dohme, Novo Nordisk, Pfizer, Sanofi; Imaging Vendors: Bruker, GE Healthcare; SMEs: Antaros, Bioxydyn, Truly; Universities: Chalmers, Dijon, Groningen, Leeds, Lund, Manchester, Nijmegen, Sheffield.

## Author Contributions

**Conceptualization:** René in 't Zandt, Marta Tibiletti, Geoff J. M. Parker, Lars E. Olsson.

**Data curation:** René in 't Zandt.

**Formal analysis:** René in 't Zandt, Marta Tibiletti.

**Funding acquisition:** Karin von Wachenfeldt, Geoff J. M. Parker, Lars E. Olsson.

**Investigation:** René in 't Zandt, Irma Mahmutovic Persson.

**Methodology:** René in 't Zandt, Irma Mahmutovic Persson, Marta Tibiletti, Karin von Wachenfeldt, Lars E. Olsson.

**Project administration:** René in 't Zandt, Irma Mahmutovic Persson, Karin von Wachenfeldt, Lars E. Olsson.

**Resources:** Irma Mahmutovic Persson, Lars E. Olsson.

**Software:** René in 't Zandt.

**Supervision:** Lars E. Olsson.

**Validation:** René in 't Zandt, Marta Tibiletti, Geoff J. M. Parker.

**Visualization:** René in 't Zandt, Irma Mahmutovic Persson.

**Writing – original draft:** René in 't Zandt.

**Writing – review & editing:** René in 't Zandt, Irma Mahmutovic Persson, Marta Tibiletti, Geoff J. M. Parker, Lars E. Olsson.

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
