## [Decision Letter · Decision Letter 0]

14 May 2024

PONE-D-23-43281Contrast enhanced longitudinal changes observed in an experimental bleomycin-induced lung fibrosis rat model by radial DCE-MRI at 9.4TPLOS ONE

Dear Dr. in 't Zandt,

Thank you for submitting your manuscript to PLOS ONE. After careful consideration, we feel that it has merit but does not fully meet PLOS ONE’s publication criteria as it currently stands. Therefore, we invite you to submit a revised version of the manuscript that addresses the points raised during the review process.

The reviewers have offered valuable feedback and proposed enhancements that could elevate the quality and lucidity of your work. They have specifically pointed out that the study design could be more transparent, particularly regarding the number of groups at various stages, which currently lacks clarity.

They have also requested additional information about the image acquisition and analysis. This encompasses the methodologies employed, the parameters established, and the interpretation of the data. Incorporating these details would provide greater context and facilitate a more comprehensive understanding of the results.

The manuscript stands to gain from a meticulous review of spelling and grammar. Such an improvement would not only enhance its readability but also ensure the effective communication of your findings. The reviewers have further suggested an upgrade in the quality of the graphs included in your paper. High-resolution and clear graphs are pivotal for presenting your data in a manner that is easily comprehensible and interpretable.

We look forward to receiving your revised manuscript.

Kind regards,

Francesca Pennati, Ph.D.

Academic Editor

PLOS ONE

Journal Requirements:

3. Thank you for stating the following financial disclosure: "TRISTAN-IHI consortium (#IB4SD-116106, https://www.imi-tristan.eu/) (Translational Imaging in Drug Safety Assessment -  Innovative Health Initiative) "  

Reviewers' comments:

Reviewer's Responses to Questions

**Comments to the Author**

1. Is the manuscript technically sound, and do the data support the conclusions?

Reviewer #1: Yes

Reviewer #2: Partly

2. Has the statistical analysis been performed appropriately and rigorously? 

Reviewer #1: Yes

Reviewer #2: Yes

3. Have the authors made all data underlying the findings in their manuscript fully available?

Reviewer #1: Yes

Reviewer #2: No

4. Is the manuscript presented in an intelligible fashion and written in standard English?

Reviewer #1: Yes

Reviewer #2: Yes

5. Review Comments to the Author

Reviewer #1: In this manuscript, René in ’t Zandt et al describe the application of DCE-MRI to a rat model of bleomycin-induced lung injury. The experiments and analyses were performed well and carefully. The work is novel, as far as I’m aware, no publication exists yet concerning DCE-MRI applied to a bleomycin model.

Major comments

-Animals: 15 rats were used in the study (line 99). In figure 2, data from 6 rats are shown as controls. Therefore, I presume that 9 rats received bleomycin. However, in figure 2, data from 6 rats were summarized at day 7 and data from 5 rats were summarized at day 28. I suppose that the same animals were measured at both time points after bleomycin. Thus, why data from only 6 respectively 5 rats were summarized instead of 9? More clarity about the experimental procedure should be provided regarding this point.

-Figure 2: Enhancement factor is by definition a factor and does not have units. Therefore, a.u. (arbitrary units) should be omitted. Also, the quality of the graphs in the pdf document was poor – increasing the font size and the quality of the graphs reproduction would be welcome.

-The colors are confusing. In fig. 2a the rectangles around the images are green, blue and purple for control, bleomycin day 7 and bleomycin day 28, respectively. However, in fig. 2b, the colors are green, purple and blue for control, bleomycin day 7 and bleomycin day 28, respectively. Please adapt the colors. In order to better discriminate control curves, I suggest plotting them with black fonts.

-Lines 272-273: The fact that the tissue remodeling detected by MRI occurs primarily around main airways has been shown earlier and confirmed histologically by comparison observations in the same regions (doi 10.1002/jmri.22476).

-Signal intensity of the tissue remodeling area around main airways, determined from images before the contrast agent injection, should also be provided. Analysis should be compared with the DCE-MRI results. It might be the case that both measures might be able to discriminate between the inflammatory and more fibrotic (tissue remodeling) phases of the bleomycin model.

-Figure 2B,C: It is not clear to me whether the curves represent the mean of an average of signal over the whole lung in every animal, or whether they are the mean of the signal over the lesions in every rat. I rather suppose that it is the first, as the mean profiles look the same in control and bleomycin animals (except obviously for the peak), which is surprising to me. I would expect the washout period in bleomycin rats to be different (longer) than in controls. I would strongly suggest to perform analyses in the areas around main airways as well, where most of the remodeling occurred, and present the results in a separate figure. Please clarify.

-Figure 3: The same comment concerning color coding made above for figure 2 holds true here.

-Figure 3B: If I understood correctly the distribution of the clusters and compare to the images shown in figure 3A, does it mean that the higher cluster volumes for lower enhancement factors at day 28 after bleomycin are related to diffuse fibrosis? This would be extremely interesting – especially when testing a therapy and comparing the results to the stronger remodeling around main airways in the central lung. Please clarify.

-Discussion: In the Introduction section, the authors mention references 8-10 summarizing clinical DCE-MRI work in IPF patients. They should briefly compare their findings with the clinical observations.

-Contrast agent: Given the fact that the use of Gd-based contrast agents in MRI examinations is being scrutinized due to reported side effects, especially in the kidney, the pros/cons of DCE-MRI in the evaluation of pulmonary fibrosis in comparison to MRI techniques not relying on the administration of contrast material should be briefly discussed.

Minor comments

-Line 74: Suggest replacing “In addition” by “Also”.

-Line 86: Suggest replacing “Therefore” by “Thus”.

-Lines 152-153 & lines 167-221: Sentences in past tense would be more appropriate.

-Results should be described in past tense (e.g. line 257).

Reviewer #2: In the manuscript the authors present their work on dynamic contrast enhanced lung MRI in a rodent model in which lung fibrosis was induced using bleomycin.

For this n = 15 rats were used, of those one group and I went a bleomycin challenge at day 0, the second group served as control and did not underwent any treatment. Lung MRI was performed at day 7 and day 28 using using a 9.4 Tesla scanner and a radial 3D UTE sequence.

Contrast agent was applied intravenously after seven minutes and dynamic contrast enhanced imaging was performed for 21 minutes. The relative enhancement was measured across the imaging time. For quantitive analysis, the enhancement intensity were binned using 6 thresholds; enhancement bins were then compared and statistically analyzed.

The main findings of this study were an increased lung volume between experimental and control group at day 28, determined by manual lung segmentation and a significantly increased DCE during inflammation stage at day 7. Interestingly that appeared not to be any different contrast enhancement in experimental group at day 28 compared to the control group.

1. Please specify the number of used rodents in the study. That included number is not given in the abstract, please add the number of overall included animals here as well as how many were used in experimental and control group. Please also add this in the method section.

I’m not sure if there is data inconsistency here: in the method section you state that n = 15 animals were included. I cannot find any information in the method section how many animals were resorted to the control and to the experimental group.

In figure 1A the n=4 control and n=12 bleomycin rats we’re included which sums in a total of 16 animals. In figure 1B it is not understandable why there are only six animals in the experimental group at a seven and five at day 28.

Please revise your numbers throughout the whole manuscript, check how many animals were used and please add this information as I requested above. Whenever there are different numbers at different time points (day 7 and day 28) please explain the missing n here!

2. Regarding the used UTE sequence: please add the radio trajectory information in the abstract as well. The TE of > 0.3 ms appears quite long here, additionally the flip angle is rather high, this is probably why you only get very low signal from the lung tissue. Was this anticipated? Is this a limit from the Bruker sequence or could TE be reduced? Please comment on this. You further state that the flip angle was set to 40° because of sensitivity to Gd contrast agent, was this done empirically or did you perform any estimations?

3. In the method section, you explained that images were acquired in coronal orientation. In the figures you show transverse images, why did you do that? Pixel resolution was anisotropic with 0.6 x 0.6 x 1.2 mm and you even zerofill to 0.3 x 0.3? This seems odd, did I miss something here? Please comment or elaborate.

4. The binning method that you used for quantification of the DCE signal is very hard to understand! This needs more explanation in the methods section and probably a graphical explanation in a figure, so that its easier to follow.

5. Please try and elaborate why there is no difference in DCE in bleomycin day 28 images. I would definitely expect extracellular accumulation of GBCA during washout phase and some kind of late enhancement when there would be fibrosis, but this cannot be seen here. Why?

Please review the manuscript regarding spelling and grammar. If possible, it should be revised by a native speaker. The manuscript contains some very complicated non idiomatic sentence structures and a few spelling errors.

6. PLOS authors have the option to publish the peer review history of their article (what does this mean?). If published, this will include your full peer review and any attached files.

Reviewer #1: **Yes: **Nicolau Beckmann

Reviewer #2: No

---

## [Author Response · Author response to Decision Letter 0]

1 Jul 2024

A fully formatted review report has been uploaded as word file as instructed. A copy of the content has been copied into this box.

Comments to Academic Editor

We have double checked the style requirements and all requirements are met as far as we can judge. 

Changes implemented:

- short title removed from title page and only mentioned in the submission system.

- reference system Vancouver style adapted from (x) to [x] according to your guidelines

- changed the level sections in the material and method section from level 3 to level 2. 

- all figures have been processed through PACE without any issues. The PDF files were visually inspected and the processed TIFF files from the PACE page were used for this revised submission. 

All material needed to reproduce the findings will be available at the Swedish National data service. A DOI will be activated upon accepting of the manuscript, and the following link will be available for review and this repository will become public on acceptance of the manuscript. 

https://doi.org/10.5878/b4cg-vc79 (not activated yet)

During the review process, the repository can be viewed using the following URL. 

https://snd.se/en/catalogue/dataset/preview/8078bcbf-8fc9-4c71-9119-eb4682d7542f/1

3. Thank you for stating the following financial disclosure: "TRISTAN-IHI consortium (#IB4SD-116106, https://www.imi-tristan.eu/) (Translational Imaging in Drug Safety Assessment - Innovative Health Initiative) " 

Please state what role the funders took in the study. If the funders had no role, please state: ""The funders had no role in study design, data collection and analysis, decision to publish, or preparation of the manuscript."" If this statement is not correct you must amend it as needed. 

We would like to use the following statement, and this is also mentioned in the cover letter. 

The research leading to these results received funding from the Innovative Medicines Initiatives 2 Joint Undertaking under grant agreement No 116106 (IB4SD-TRISTAN). This Joint Undertaking receives support from the European Union's Horizon 2020 research and innovation programme and EFPIA. The funders had no role in study design, data collection and analysis, decision to publish, or preparation of the manuscript

Reviewers' comments: Reviewer #1

In this manuscript, René in ’t Zandt et al describe the application of DCE-MRI to a rat model of bleomycin-induced lung injury. The experiments and analyses were performed well and carefully. The work is novel, as far as I’m aware, no publication exists yet concerning DCE-MRI applied to a bleomycin model.

Major comments

-Animals: 15 rats were used in the study (line 99). In figure 2, data from 6 rats are shown as controls. Therefore, I presume that 9 rats received bleomycin. However, in figure 2, data from 6 rats were summarized at day 7 and data from 5 rats were summarized at day 28. I suppose that the same animals were measured at both time points after bleomycin. Thus, why data from only 6 respectively 5 rats were summarized instead of 9? More clarity about the experimental procedure should be provided regarding this point.

The following was added to the material and methods to clarify the animal group sizes.

A total of 15 animals were included in this study. 4 animals were used as controls (receiving only Saline instillation) and 11 animals received bleomycin. Based on prior experiences with the bleomycin model, 5 out of 11 animals were designated as backup subjects from the imaging group of bleomycin exposed animals, to maintain the study's reliability and continuity in case of unexpected issues with the primary subjects, such as extensive weight loss or imaging or anesthesia related events. Day 7 and day 28 were chosen for DCE-MRI scan days as two representative days of dominant inflammatory and fibrotic time points respectively. The planned group size for DCE-MRI in bleomycin exposed animals was n=6.

We also added the following section to the Results section.

DCE-MRI was successful in 3 out of 4 control animals on day 7, for 1 animal the Venflon catheter punctured the venous artery and substantial amount of fluid was injected extra-vascular in the tail. This animal was removed from the DCE-MRI on day 28 due to vascular and tissue damage in the tail. Subsequently, only 3 controls were available for DCE-MRI on day 28 as well. In the bleomycin-exposed group, all DCE-MRI injections (n=6) were completed successfully on day 7. On day 28, one contrast agent injection partly failed as the images showed that only a fraction of the contrast agent reached the lung. This reduced the group size for DCE-MRI in bleomycin-exposed group animals to n=5.

-Figure 2: Enhancement factor is by definition a factor and does not have units. Therefore, a.u. (arbitrary units) should be omitted. Also, the quality of the graphs in the pdf document was poor – increasing the font size and the quality of the graphs reproduction would be welcome.

Thank you for this comment. The authors agree on this point and have removed this label from the figures when enhancement factor was plotted on the y-axis. 

We agree that the quality of the graphs in the pdf document provided for review is poor. The link provided on the right top corner on the appropriate page in the submission pdf file we have, contains links to the tiff version of the files. These files have the quality required by PLOS ONE as far as we can judge. For the resubmission, we have used the PACE diagnostic tool as recommend by PLOS ONE to make sure the quality requirements are met in this revision. 

-The colors are confusing. In fig. 2a the rectangles around the images are green, blue and purple for control, bleomycin day 7 and bleomycin day 28, respectively. However, in fig. 2b, the colors are green, purple and blue for control, bleomycin day 7 and bleomycin day 28, respectively. Please adapt the colors. In order to better discriminate control curves, I suggest plotting them with black fonts.

Thank you for this notification. Unfortunately, the rectangular color boxes surrounding the images in figures 2 and 3 were switched for animals measured on day 7 and day 28. We have fixed this and apologize for this confusion. 

-Lines 272-273: The fact that the tissue remodeling detected by MRI occurs primarily around main airways has been shown earlier and confirmed histologically by comparison observations in the same regions (doi 10.1002/jmri.22476).

This is a good point; we have mentioned this in the manuscript and included this reference

-Signal intensity of the tissue remodeling area around main airways, determined from images before the contrast agent injection, should also be provided. Analysis should be compared with the DCE-MRI results. It might be the case that both measures might be able to discriminate between the inflammatory and more fibrotic (tissue remodeling) phases of the bleomycin model.

Thank you for this comment and we realize that adding the tissue remodeling/lesion area would improve our work. We lack the same gradient echo sequence used in previous publications to determine these areas in the lung. Therefore, we reconstructed a new baseline image that uses the data acquired over 5m40s prior to the injection of the contrast agent. The histogram analyses were performed like what has been described previously by our group. The volume of the pixels with an intensity higher than the intersection with the x-axis are calculated, plotted and added as figure 1C in the manuscript. The pixels showing the area of this ‘lesion/tissue remodeling’ have been added to figure 3 as an extra overlay to the MRI images. 

We have not been able to reveal any correlation between the tissue remodeling and lesions area neither have we been able to see any changes in the washout rate for the binned DCE curves. 

-Figure 2B,C: It is not clear to me whether the curves represent the mean of an average of signal over the whole lung in every animal, or whether they are the mean of the signal over the lesions in every rat. I rather suppose that it is the first, as the mean profiles look the same in control and bleomycin animals (except obviously for the peak), which is surprising to me. I would expect the washout period in bleomycin rats to be different (longer) than in controls. I would strongly suggest to perform analyses in the areas around main airways as well, where most of the remodeling occurred, and present the results in a separate figure. Please clarify.

The curves shown in figure 2B represent the average pixel intensity curves over the total lung. We have added ‘total’ to the text to emphasize that. 

The washout period being longer in bleomycin rats was also one of our assumptions to prove. We can only conclude that based on our data we cannot show that. As explained above, the binning approach didn’t reveal any difference in the dynamics. But these are the dynamics for pixels without any connection to its anatomical location in the lung. 

Looking at the volume that each bin class represents in the lung, a clear difference can be observed for the bleomycin-exposed animals. Interestingly, this difference was found for the lower signal intensity pixels rather than the high intensity pixels who correspond to the lesions determined by the histogram analysis. Pixels with lower signal intensity are mostly found on the lateral side of the lung lobes and, although it would be tempting to attribute this to collagen formation, the location doesn’t match the presumptive dominant lesion area and it is more likely that “simple expanded lung volume” with reduced tissue density is the main explanation for the finding observed. This is also what we argue for in our manuscript. 

Analysis close to the major airways is a challenge for the DCE experiment due to the high signal intensity of the vessels and the need to analyze the dynamics in tissue in proximity with these vessels and airways. The vessels pulsate with the heartbeat and the respiration creates substantial movement of tissue. The data shown is not corrected for either of these physiological processes and signal bleed is to be expected to play a major role

-Figure 3: The same comment concerning color coding made above for figure 2 holds true here.

The authors agree and the error has been fixed

-Figure 3B: If I understood correctly the distribution of the clusters and compare to the images shown in figure 3A, does it mean that the higher cluster volumes for lower enhancement factors at day 28 after bleomycin are related to diffuse fibrosis? This would be extremely interesting – especially when testing a therapy and comparing the results to the stronger remodeling around main airways in the central lung. Please clarify.

This would be the preferred conclusion, but we have no real proof of that. It is more likely this is correlated with the increased lung volume, a decrease in cell density and therefore an apparent lower MRI signal. Histology supports this observation. We opted to go for the most likely explanation based on the lung volume differences found. 

-Discussion: In the Introduction section, the authors mention references 8-10 summarizing clinical DCE-MRI work in IPF patients. They should briefly compare their findings with the clinical observations.

We agree and have modified a section in the discussion to reconnect to that section in the introduction, with the modification underlined.

….Substantial inflow effects and the relative low time resolution compared to the mean transit time in the rat lung [11], hampers a model-based analysis in our experiments. Additionally, these factors complicate the direct translation of perfusion-sensitive DCE-MRI used clinically, as this application relies on the measurement of the first-passage characteristics of the injected contrast agent [9, 10]. It cannot be excluded that using a scanning DCE approach with significantly higher temporal and spatial resolution more details in the bleomycin model could be revealed…..

-Contrast agent: Given the fact that the use of Gd-based contrast agents in MRI examinations is being scrutinized due to reported side effects, especially in the kidney, the pros/cons of DCE-MRI in the evaluation of pulmonary fibrosis in comparison to MRI techniques not relying on the administration of contrast material should be briefly discussed.

Patients with idiopathic pulmonary fibrosis are known to have a higher prevalence of chronic kidney disease, which is also regarded as a comorbidity. The prevalence of chronic kidney disease in this patient group increases with age, mirroring the age-related increase seen in patients with idiopathic pulmonary fibrosis itself. In a clinical setting, patients referred to an MRI-examination are checked for potential contraindications, e.g. contraindications to receiving gadolinium-based contrast agents, which potentially can cause nephrogenic systemic fibrosis. For patients with known or suspected kidney disease a blood test is performed to measure the level of creatinine and to calculate the estimated glomerular filtration rate. Depending on the result the patient may or may not undergo the MRI-examination with a gadolinium-based contrast agent. 

The above discussion is well-known and of course important for clinical application of DCE-MRI to IPF-patients. Our paper focuses on the translational capability of DCE-MRI to a preclinical setting, and we argue that the above discussion is beyond our scope.

Minor comments

-Line 74: Suggest replacing “In addition” by “Also”. 

We have adapted this.

-Line 86: Suggest replacing “Therefore” by “Thus”. 

We have adapted this.

-Lines 152-153 & lines 167-221: Sentences in past tense would be more appropriate

We have adapted this.

-Results should be described in past tense (e.g. line 257). 

Thank you for pointing out this inconsistency, we have checked that all is written in the past tense throughout the results section and further checked the manuscript on tense inconsistencies

Reviewers' comments: Reviewer #2

In the manuscript the authors present their work on dynamic contrast enhanced lung MRI in a rodent model in which lung fibrosis was induced using bleomycin.

For this n = 15 rats were used, of those one group and I went a bleomycin challenge at day 0, the second group served as control and did not underwent any treatment. Lung MRI was performed at day 7 and day 28 using using a 9.4 Tesla scanner and a radial 3D UTE sequence.

Contrast agent was applied intravenously after seven minutes and dynamic contrast enhanced imaging was performed for 21 minutes. The relative enhancement was measured across the imaging time. For quantitive analysis, the enhancement intensity were binned using 6 thresholds; enhancement bins were then compared and statistically analyzed.

The main findings of this study were an increased lung volume between experimental and control group at day 28, determined by manual lung segmentation and a significantly increased DCE during inflammation stage at day 7. Interestingly that appeared not to be any different contrast enhancement in experimental group at day 28 compared to the control group.

1. Please specify the number of used rodents in the study. That included number 

---

## [Decision Letter · Decision Letter 1]

4 Sep 2024

Contrast enhanced longitudinal changes observed in an experimental bleomycin-induced lung fibrosis rat model by radial DCE-MRI at 9.4T

PONE-D-23-43281R1

Dear Dr. Zandt,

We’re pleased to inform you that your manuscript has been judged scientifically suitable for publication and will be formally accepted for publication once it meets all outstanding technical requirements.

Kind regards,

Minghua Wu, M.D., Ph.D.

Academic Editor

PLOS ONE

Additional Editor Comments (optional):

Reviewers' comments:

Reviewer's Responses to Questions

**Comments to the Author**

1. If the authors have adequately addressed your comments raised in a previous round of review and you feel that this manuscript is now acceptable for publication, you may indicate that here to bypass the “Comments to the Author” section, enter your conflict of interest statement in the “Confidential to Editor” section, and submit your "Accept" recommendation.

Reviewer #1: All comments have been addressed

2. Is the manuscript technically sound, and do the data support the conclusions?

Reviewer #1: Yes

3. Has the statistical analysis been performed appropriately and rigorously? 

Reviewer #1: Yes

4. Have the authors made all data underlying the findings in their manuscript fully available?

Reviewer #1: Yes

5. Is the manuscript presented in an intelligible fashion and written in standard English?

Reviewer #1: Yes

6. Review Comments to the Author

Reviewer #1: All points that I raised have been carefully addressed by the authors in this revised version. Manuscript became clearer.

7. PLOS authors have the option to publish the peer review history of their article (what does this mean?). If published, this will include your full peer review and any attached files.

Reviewer #1: **Yes: **Nicolau Beckmann

---

## [Editor Report · Acceptance letter]

19 Sep 2024

PONE-D-23-43281R1 

PLOS ONE

Dear Dr. in 't Zandt, 

I'm pleased to inform you that your manuscript has been deemed suitable for publication in PLOS ONE. Congratulations! Your manuscript is now being handed over to our production team.

Kind regards, 

on behalf of

Dr. Minghua Wu 

Academic Editor

PLOS ONE